# Phosphorylation of Microglial IRF5 and IRF4 by IRAK4 Regulates Inflammatory Responses to Ischemia

**DOI:** 10.3390/cells10020276

**Published:** 2021-01-30

**Authors:** Conelius Ngwa, Abdullah Al Mamun, Yan Xu, Romana Sharmeen, Fudong Liu

**Affiliations:** Department of Neurology, McGovern Medical School, The University of Texas Health Science Center at Houston, Houston, TX 77030, USA; Conelius.Ngwa@uth.tmc.edu (C.N.); Abdullah.a.Mamun@uth.tmc.edu (A.A.M.); Yan.Xu.1@uth.tmc.edu (Y.X.); Romana.Sharmeen@uth.tmc.edu (R.S.)

**Keywords:** inflammation, IRAK4, IRF5, IRF4, ischemia, microglia

## Abstract

Background: Interferon Regulatory Factor (IRF) 5 and 4 play a determinant role in regulating microglial pro- and anti-inflammatory responses to cerebral ischemia. How microglial IRF5 and IRF4 signaling are activated has been elusive. We hypothesized that interleukin-1 receptor associated kinase 4 (IRAK4) phosphorylates and activates IRF5 and IRF4 in ischemic microglia. We aimed to explore the upstream signals of the two IRFs, and to determine how the IRAK4-IRF signaling regulates the expression of inflammatory mediators, and impacts neuropathology. Methods: Spontaneously Immortalized Murine (SIM)-A9 microglial cell line, primary microglia and neurons from C57BL/6 WT mice were cultured and exposed to oxygen-glucose deprivation (OGD), followed by stimulation with LPS or IL-4. An IRAK4 inhibitor (ND2158) was used to examine IRAK4′s effects on the phosphorylation of IRF5/IRF4 and the impacts on neuronal morphology by co-immunoprecipitation (Co-IP)/Western blot, ELISA, and immunofluorescence assays. Results: We confirmed that IRAK4 formed a Myddosome with MyD88/IRF5/IRF4, and phosphorylated both IRFs, which subsequently translocated into the nucleus. Inhibition of IRAK4 phosphorylation quenched microglial pro-inflammatory response primarily, and increased neuronal viability and neurite lengths after ischemia. Conclusions: IRAK4 signaling is critical for microglial inflammatory responses and a potential therapeutic target for neuroinflammatory diseases including cerebral ischemia.

## 1. Introduction

Immune responses are a fundamental pathophysiological procedure in cerebral ischemia [1], and microglial activation plays a central role in initiating and perpetuating the inflammatory response [2]. After cerebral ischemia, microglia are activated producing and releasing a plethora of cytokines, chemokines, cytotoxic mediators and trophic factors, which consequently mediate pro- and anti-inflammatory responses [3,4,5,6,7]. Microglia with the pro-inflammatory phenotype are associated with detrimental outcomes [8,9]. In contrast, microglia of the anti-inflammatory phenotype promote tissue repair and confer neuroprotective effects [10,11,12,13]. Over-activation of microglia exacerbates cerebral ischemic injury [11,14,15]; therefore, it has high translational value to modulate microglial activation via inhibiting or skewing the pro-inflammatory to the anti-inflammatory phenotype [11,12].

Our previous in vivo studies using genetic manipulations in animals have found interferon regulatory factor 5 (IRF5) and 4 (IRF4) play critical roles in mediating microglial pro- and anti-inflammatory response respectively after stroke. However, through which pathway IRF5/IRF4 mediate the microglial response remains elusive. Systemic inflammation studies on macrophage activation have suggested interleukin-1 receptor-associated kinase 4 (IRAK-4) and adaptor protein MyD88 are crucial in phosphorylation of IRFs [14,15,16,17,18,19,20]. We hypothesize that the IRAK4-IRF signaling phosphorylates the two IRFs in microglia, and facilitates IRFs’ translocation from the cytoplasm to the nucleus to regulate gene expression of inflammatory mediators. In this study, we specifically performed a series of in vitro assays in microglia culture exposed to oxygen glucose deprivation (OGD), an in vitro ischemia model, to elucidate the molecular mechanism. The interaction of IRF5/IRF4 with MyD88 and IRAK4 was examined. The effects of IRAK4 inhibition on the activation of the two IRFs and the impacts on ischemic neurons were determined.

## 2. Materials and Methods

### 2.1. Spontaneously Immortalized Murine Microglia Culture

Spontaneously immortalized murine microglia, SIM-A9_CRL-3265 (American Type Culture Collection, ATCC, Manassas, VA, USA) cells were cultured as previously described [21]. Briefly cells were cultured in DMEM/F-12 medium_DFL15 (Caisson Laboratories, Smithfield, UT, USA) containing 10% FBS, 5% HS, and 1% P/S. Next, cells were gently shaken and detached from the culture vessel with phosphate buffered saline (PBS) containing 1 mM EDTA, 1 mM EGTA and 1 mg/mL D-glucose. The detached cells were counted with a hemocytometer, and rested for at least 24 h in a Forma Steri-Cycle CO_2_ incubator (Thermo Scientific, Waltham, MA, USA) at 37 °C, 95% humidity and 5% CO_2_, before treatments. The cell line can yield more than 5 million cells after culture and was used for western blotting experiments (Figure 1, Figure 2 and Figure 3).

### 2.2. Primary Microglia Culture

Primary microglia were obtained from breeding colonies kept in barrier-reared conditions in a pathogen-free facility at the University of Texas Health Science Center at Houston, by following Draheim’s method [22] with modifications. Briefly, brains from post-natal day 0–4, C57BL/6 WT mice were excised in an aseptic environment, and the cortices freed from the midbrain and meninges, and the cells isolated using neural tissue dissociation kit P_130092628 (Miltenyi Biotech, Auburn, CA, USA). The cells were cultured in Dulbecco’s Modified Eagle Medium (DMEM) containing 10% FBS and 1% P/S, and in poly-D-lysine (PDL)-coated T75 flasks. After 24 h, the medium was replaced with fresh DMEM containing FBS and P/S. The culture was boosted with 6% L929-conditioned medium at day 3–5. After 12 days, flasks were shaken for 2–3 h at 250 RPM and at 37 °C, to collect loosely attached microglia. The purity of these microglial cells was 99% as determined by Iba-1 immunoreactivity in Leica DMi8 Confocal Microscope (Figure 5B).

### 2.3. Cortical Neurons

Primary neurons were prepared as previously reported [23] with modifications. Cortices from embryos of E16–18 were isolated and immersed in cold DMEM_11965118. The DMEM in the cortices (from 4 embryos) was removed by centrifugation at 250 RPM for 1 min, followed by incubation of the cortices in 1 mL 0.25% trypsin/EDTA_25200056 (Thermo Fisher Scientific, Waltham, MA, USA) at 37 °C for 15 min. The cortices were next triturated (×10) and incubated for another 15 min. Trypsin/EDTA was quenched with BSA (0.5 mL) and the cell suspension was gently homogenized to clear clog formed. Single cells were obtained by passing the cell suspension through a 40–70 µm cell strainer. We plated 200,000 mixed cells per well of PDL-coated 8-chamber slide, and then incubated cells in a Steri-Cycle CO_2_ incubator (Forma, Marietta, OH, USA) at 37 °C, 95% humidity and 5% CO_2_, for 24 h. DMEM supplemented with 10% FBS and 1% P/S was used for the initial plating (24 h), and then the media was replaced with neurobasal_21103049 containing 2% B-27™ supplement_17504044 and 1% P/S. Half the media was replaced every 3 days and completely replaced at division day 6 (DIV 6) and DIV 12. After 15 days, cells were ready for treatments. Neurite formation and Microtubule Associated Protein 2 (MAP2) immune reactivity were examined with a DMi8 Confocal Microscope (Leica, Buffalo Grove, IL, USA) and analyzed with ImageJ (NIH, Bethesda, MD, USA).

### 2.4. ND2158 Treatment, Oxygen-Glucose Deprivation (OGD), and LPS or IL-4 Stimulation

SIM-A9 or primary microglia were treated with 8 µM ND2158 during the OGD and reperfusion period. OGD was performed as previously reported [24,25,26], with modifications. Briefly, the culture medium was replaced with glucose-free DMEM_A1443001 (Thermo Fisher Scientific), and then a sealed chamber was used to place the plates (opened), followed by oxygen expiration for 10 min via flowing in 95% N_2_ and 5% CO_2_ mixture persistently. The chamber was transferred into a 37 °C incubator after clamping the inlet and outlet for 4 h. In the course of OGD, O_2_ levels dropped to <2% after 2 h, and to <1% at 4 h, as shown by a change in color of BD_271051 anaerobic indicator strips (inserted both inside and outside of the chamber) from blue (aerobic) to white (anaerobic) coloration. After 4 h OGD, microglia were reperfused by replacement with new medium containing normal levels of glucose plus LPS or recombinant mouse IL-4 protein_404-ML010 (100 ng/mL), followed by cell incubation in a Forma Steri-Cycle CO_2_ incubator at 37 °C, 95% humidity and 5% CO_2,_ for 24 h. As neurons are less resistant to OGD than microglia, neuronal OGD was performed for 2 h; and then neurons were reperfused with the medium collected from the treated microglia culture, followed by incubation at 37 °C, 95% humidity and 5% CO_2,_ for 24 h. We plated SIM-A9 cells at a density of 0.5 × 10^6^/well of 6-well plate until 75–80% confluence was achieved; meanwhile the primary microglia were plated at a density of 1.5 × 10^6^/well. All experiments were performed paralleled with normoxic controls, and in triplicates.

### 2.5. Immunocytochemistry (ICC)

ICC was performed as previously described [9]. The following primary antibodies were used: IRF5 (10T1) sc-56714, 1:100; IRF4 (F-4) sc-48338, 1:100 (Santa Cruz Biotechnology Inc., Dallas, TX, USA), anti Iba-1_01919741, 1:200 (FUJIFILM Wako Chemical Corporation, Richmond, VA, USA), MAP2_PA517646, 1:200 (Thermo Fisher Scientific). The fluorophore-conjugated secondary antibodies used at 1:400 included Donkey anti-mouse Alexa Fluor 488_A-32766, Donkey anti-Rabbit Alexa Fluor 594_A-21207, and Donkey anti-Rabbit Alexa Fluor 488_R37118 (Thermo Fisher Scientific). Images were captured in Leica DMi8 Confocal Microscope, and fluorescence intensities were quantified by ImageJ.

### 2.6. Co-Immunoprecipitation (Co-IP)

Protein was extracted based on Bohgaki’s method [27] and quantified with Pierce BCA protein assay kit_23225. Primary antibodies (15 μg) or IgG negative control was added to the crude protein (0.5–1 mg/mL per experiment) in PBST, and then the mixture was incubated for 1 h at 2–8 °C, with tilting and rotation. 40 μL protein G PLUS-Agarose_sc-2002 (Santa Cruz Biotechnology Inc. Dallas, TX, USA) was washed with PBST, and then incubated with the protein-antibody complex at 4 °C for 24 h. The protein G PLUS-Agarose-antibody-antigen complex was washed again with PBST and the complex collected after centrifugation at 4 °C and at 2500 RPM, RT for 5 min. The protein was eluted with sample buffer supplemented with 2-mercaptoethanol_M3148 (Sigma-Aldrich, St. Louis, MO, USA), and then centrifuged at 2500 RPM, RT, for 5 min, before SDS-PAGE/Western blotting. The antibodies used for Co-IP included normal IgG_sc-2025, MyD88 (E-11)_sc-74532, IRF5 (10T1)_sc-56714, IRF4 (F-4)_sc-48338 (Santa Cruz Biotechnology Inc. Dallas, TX, USA); normal IgG_2729S, IRAK4_4363 (Cell Signaling Technology, Danvers, MA, USA); IRAK4 (2H9)_MA5-15883 (Thermo Fisher Scientific).

### 2.7. Cell Fractionation

We used NE-PER Nuclear and Cytoplasmic Extraction reagent_78835 (Thermo Fisher Scientific) to obtain cytoplasmic and nuclear protein fractions for both treated and control microglial cells, based on the vendor’s instructions.

### 2.8. Western Blotting

Proteins extracted from cells or isolated by Co-IP (up to 50 µg) were separated in 4–15% Mini-PROTEAN™ TGX Gels (BioRad, Hercules, CA, USA) at 75–120 V for 1.5 h, and then transferred onto a nitrocellulose membrane at 150 mA for 2 h. Immunoblotted proteins on the membrane, labelled with specific antibodies were imaged by autoradiography and signals quantified by ImageJ. All primary and secondary antibodies were used according to the manufacturer’s instructions. The primary antibodies used were: MyD88 (D80F5)_4283 1:1000, β-actin_4970 1:1000, IRAK4_4363 1:1000, Phospho-IRAK4 (Thr345/Ser346) (D6D7)_11927 1:1000 (Cell Signaling Technology, Danvers, MA, USA). Anti-beta tubulin_ab6046 1:1000 (abcam, Cambridge, MA, USA). IRF5_PA5-19504 1:1000, IRF4_PA5-21144 1:1000, Phospho-IRF5 (Ser437)_PA5-106093 1:500, IRAK4 (2H9)_MA5-15883 1:1000 (Thermo Fisher Scientific); IRF4 (Phospho-Tyr122/125) _2846 1:500 (SAB Signaling Antibody, College Park, MD), MyD88 (E-11)_sc-74532 1:500, IRF5 (10T1)_sc-56714 1:500, IRF4 (F-4)_sc-48338 1:500, Lamin B1(B-10)_sc-374015 1:500 (Santa Cruz Biotechnology Inc.). The secondary antibodies were: peroxidase_PI-1000 1:4000, and peroxidase_PI-2000 1:4000 (VECTOR Laboratories, Burlingame, CA, USA).

### 2.9. Enzyme-Linked Immunosorbent Assay (ELISA)

Cell culture medium was collected and tested for levels of TNF-α, IL-6, TGF-β1 and Arg-1 by using mouse ELISA MAX Deluxe or LEGEND MAX (TNF-α_430904, IL-6_431304, and Total TGF-β1_436707 (BioLegend, San Diego, CA, USA), mouse Arg-1_LS-F6864 (LifeSpan BioSciences, Inc, Seattle, WA, USA), according to the manufacturer’s instructions. Signals were measured at 450 nm in EnSpire^TM^ Multimode Plate Reader (Perkin Elmer, Inc., Richmond, CA, USA).

### 2.10. mRNA Extraction and Real-Time Polymerase Chain Reaction (RT-PCR)

Total RNA was extracted using RNeasy Mini Kit_74104 (QIAGEN, Germantown, MD, USA) according to the manufacturer’s protocol, and quantified using NanoDrop (Thermo Fisher Scientific). The RNA was converted to cDNA by iScript™ Reverse Transcription Supermix_1708841. C1000 Touch Thermal Cycler CFX96 Real-Time System (Bio-Rad, Hercules, CA, USA) and the SsoAdvanced Universal SYBR Green Supermix_1725274 (Bio-Rad) were used to perform Q-PCR. The following gene primers from Integrated DNA Technologies (Coralville, IA, USA) were used: Ym-1/2 F_CAGGGTAATGAGTGGGTTGG, Ym1/2 R_CACGGCACCTCCTAAATTGT; and the housekeeping gene GAPDH F_GTGTTCCTACCCCCAATGTGT, GAPDH R_ ATTGTCATACCAGGAAATGAGCTT. The results are reported as normalized fold changes in mRNA, which were determined via the ΔΔCt method using the threshold cycle (Ct) value for gene of interest.

### 2.11. Calcein Cell Viability Assay

Calcein AM cell viability/cytotoxicity Assay kit_4892-010-K (R&D Systems, Inc., Minneapolis, MN, USA) was used to examine neuronal viability. Briefly, PDL-coated 96-well tissue culture plates with cultured neurons (50,000/well) were exposed to OGD for 2 h, followed by re-perfusion with media from treated microglia for 24 h. Next the neurons were incubated in 1X Calcein AM DW buffer containing Calcein AM (1 µM, 100 µL) for 30 min in a Forma Steri-Cycle CO_2_ incubator at 37 °C, 95% humidity and 5% CO_2_. Relative fluorescence unit (RFU) readings were obtained in an EnSpire^TM^ Multimode Plate Reader (Perkin Elmer, Inc., Richmond, CA, USA) using 485 nm excitation filter and 520 nm emission filter. The fluorescence intensity was proportional to the number of intact viable cells easily permeated by Calcein AM (a non-fluorescent, hydrophobic compound) and hydrolyzed intracellularly to calcein (fluorescent, hydrophilic compound). Experiments were performed in replicates (*n* = 4–6).

### 2.12. Statistical Analysis

Statistical data analysis was performed using Prism 8.0.2 (GraphPad Software, San Diego, CA, USA) with *p* < 0.05 considered statistically significant. All data are presented as the mean ± standard error of the mean (SEM), and analyzed using one-way ANOVA with Tukey post-hoc test for multiple comparisons. All *n* and *p* values and statistical tests are indicated in figure legends.

## 3. Results

### 3.1. Interaction of IRF5/IRF4 with MyD88/IRAK4

We have previously reported the expression of IRF5 and IRF4 in microglia [9,28]. IRAK4 and MyD88 are critical for IRF5 phosphorylation in peripheral macrophages [14,16,29,30,31]. Here we examined if microglial IRF5 and/or IRF4 also interacts with IRAK4 and/or MyD88 in an in vitro assay with SIM-A9 microglia culture. To mimic the in vivo ischemic condition, we exposed the microglial culture to a 4 h OGD [24,25,26] followed by LPS (pro-inflammatory) or IL-4 (anti-inflammatory) stimulation [9,32,33,34,35,36,37,38,39,40]. One day after the stimulation, cell homogenates were subjected to co-immunoprecipitation (Co-IP) assays. As shown in Figure 1A,B, we found strong signals of either total IRF5/4 (t-IRF5/4) or phosphorylated IRF5/4 (p-IRF5/4) in the compounds precipitated by IRAK4 antibody (IRAK4 IP) in each treatment, although relatively weaker signals were seen in unstimulated controls (the last lane to the right of each blot). Similarly, t-IRF5/4 and p-IRF5/4 were also seen in MyD88-precipitated (MyD88 IP) compounds (Figure 1E,F). MyD88 was also detected in the IRAK4 precipitation, suggesting a Myddosome [16,22,41,42] formed for IRF phosphorylation (Figure 1A,B). In each Co-IP, the phosphorylated IRFs were quantified as the ratio over the total IRFs, and there were no significant differences in the ratios between treatments. These Co-IP data indicated direct interactions between IRAK4, MyD88 and IRF5/4, suggesting IRAK4 and MyD88 are closely involved in IRF5/4 phosphorylation in microglia.

### 3.2. IRAK4 Phosphorylation Triggers Phosphorylation of IRF5 and IRF4

To test if IRAK4 plays a critical role in the phosphorylation of IRF5/IRF4, we performed culture of SIM-A9 microglia cell line [21], and subjected the culture to a 4 h OGD, in the presence of 8 µM ND2158 (inhibitor of IRAK4) [43]. The dose 8 μM ND2158 was selected based on the viability gradient assay by CellTiter 96^®^ AQueous Non-Radioactive Cell Proliferation Assay [41,44] that identified the largest dose with above 75% cell viability (Figure 2A).

Next, we performed western blotting by using a highly specific antibody to detect phosphorylation of IRAK4 at Thr-345 and Ser-346 [31] in the cell homogenates, and probed for both total and phosphorylated forms of IRF5/IRF4 (Figure 2B–E). As expected, ND2158 treated cells showed nearly null expression of p-IRAK4 after stimulation with OGD + LPS or +IL-4, indicating a complete inhibition of the kinase (Figure 2B,C). Interestingly, the exactly same pattern was seen in the expression of p-IRF5 or p-IRF4; suggesting the phosphorylation of both IRFs were also blocked by the IRAK4 inhibitor (Figure 2B,D,E). In the ND2158 untreated cells (Figure 2B, lanes 2&4), phosphorylated forms of IRAK4, IRF5/IRF4 were strongly detected after the stimulation, and quantification data showed the levels of p-IRAK4, p-IRF5, and p-IRF4 were significantly higher than that in ND2158 treated cells (Figure 2C–E). Phosphorylation of these proteins were also seen in control cells (Figure 2B, lane 5 from the left of each phosphorylation blot), reflecting a low baseline level of phosphorylation of these inflammatory mediators in normal SIM-A9 microglia. These data suggested that IRAK4 phosphorylates both IRF5 and IRF4 and leads to microglial pro- or anti-inflammatory response depending on the pathogenic stimuli the cell receive.

### 3.3. Phosphorylated IRF5 and IRF4 Translocate from the Cytoplasm to the Nucleus

It is a common mechanism for transcription factors to be phosphorylated and then translocated from the cytoplasm to the nucleus [44,45,46,47]. To confirm the nuclear translocation of IRF5 and IRF4 in microglia, we first stimulated cultured microglia (SIM-A9) with OGD + LPS or + IL-4, and then fractionated the cell homogenates into cytosolic and nuclear fractions. Western blotting was performed in each fraction to examine p-IRF5 or p-IRF4 protein levels. Lamin b1 and β-tubulin were used to verify the purity of cytosol and nuclear fraction respectively. As expected, the nuclear fraction showed expression of p-IRF5 or p-IRF4 under each treatment combination, indicating the nuclear translocation of both phosphorylated transcription factors (Figure 3A,B). To quantify the translocation level of these IRFs, we first normalized the optical density of the western blotting bands to the loading control β-actin, and then calculated the ratio of p-IRFs in the nucleus over cytoplasm. For p-IRF5, the ratio is significantly higher in OGD + LPS treated microglia compared to either normoxia or OGD alone control (Figure 3C); and similarly for p-IRF4, OGD + IL-4 treated cells had significantly higher ratio than both the controls (Figure 3B,D). Interestingly, OGD + IL4 and OGD + LPS had minimal effect on p-IRF5 and p-IRF4 translocation respectively. These data were consistent with the notion that LPS stimulation favors microglial pro-inflammatory response (higher level of p-IRF5 translocation), and IL-4 favors the anti-inflammatory response (higher level of p-IRF4 translocation).

To further confirm the nuclear translocation of the IRFs, immunocytochemistry (ICC) was performed in cultured SIM-A9 cells (Figure 3E,F). As shown in the upper panels of both Figure 3E,F, the IRF signal (green) was absent in the nuclear region in the normoxia treated cells. However, under either OGD + LPS or OGD + IL-4 condition, the IRF signal were robustly seen in the nuclear area, suggesting significant translocations of these IRFs after pathogenic stimulation.

### 3.4. IRAK4 Modulates Microglial Activation

We next analyzed the functionality of IRAK4 on IRF5/IRF4 activation by analyzing microglial responses and detected the levels of both pro- and anti-inflammatory mediators in primary microglia culture with ELISA and RT-PCR. The result showed OGD + LPS induces robust expression of pro-inflammatory cytokines (TNF-α and IL-6) in the ND2158 absent medium; while the inhibitor or OGD + IL-4 treated cells produced very low levels of TNF-α/IL-6 (Figure 4A,B). The anti-inflammatory cytokine IL-10 level was increased by both LPS and IL-4 stimulation, and ND2158 treatment blocked the increase only in the OGD + LPS group (Figure 4C). We performed RT-PCR in cell lysates to detect mRNA of another anti-inflammatory mediator Ym1/2 (not detectable by ELISA), and found a robust increase with OGD + IL-4 stimulation, which was also attenuated by ND2158 (Figure 4D).

The data indicated that the activation of IRAK4 influences the production of both pro- and anti-inflammatory mediators, with a more robust effect on the pro-inflammatory cytokines.

### 3.5. Microglial Culture Medium Conditioned by ND2158 Increases Neuronal Viability

Our data suggested an important role of IRAK4 in producing inflammatory cytokines in microglia (Figure 4). Next, we tested whether this effect impacts neuronal morphological integrity and viability. First, the primary microglial culture was subjected to ND2158 and OGD + LPS or +IL-4 treatment. Next, the conditioned medium from the microglia culture was added to primary neuronal culture immediately after the neurons were subjected to 2 h OGD. The addition of inflammatory medium OGD + LPS or OGD + IL-4 both resulted in significantly lower calcein RFU compared with the untreated normoxia control; however, the media with addition of ND2158 reversed the RFU (Figure 5C), suggesting the inhibition of IRAK4 can increase neuronal survival in ischemia. To confirm the result, we further examined the morphological changes of neurons by ICC (Figure 5D), and found similar effects of ND2158 on neuronal MAP2 fluorescence intensity (Figure 5E) and average neurite length (Figure 5F) as that in the neuronal viability assay. There was no difference in calcein RFU between OGD alone group and the empty control, although the OGD alone treatment led to compromised MAP2 intensity and neurite length. Taken together, these data indicate that IRAK4 inhibition confers neuroprotection against ischemic injury.

## 4. Discussion

Microglia are innate immune responders and play a critical role in the progress of cerebral ischemic injury [48,49]. The interferon (IFN) regulatory factor (IRF) family were originally identified as transcription factors of type I IFN including nine different forms from IRF1 to IRF9 based on their binding motifs [50]. Among these IRFs, IRF5 and IRF4 are the key determinants in regulating microglial pro- and anti-inflammatory polarization respectively [9,51]. However, the molecular mechanisms through which microglial IRF5/4 are not clear. In the present study, we showed that microglial IRF5 and IRF4 are both phosphorylated by IRAK4 upon inflammatory stimulation. In addition, we demonstrated that the phosphorylated IRF5/IRF4 translocate from the cytoplasm to the nucleus under pathological conditions. We further found that the IRAK4 inhibition by ND2158 has a more robust effect on the production of pro-inflammatory cytokines and confers neuroprotection. To our knowledge, this is the first study that reported how microglial IRF5/4 are phosphorylated/activated, and that the translocation of p-IRFs from the cytoplasm to the nucleus is essential for microglial production of inflammatory mediators.

Our previous work [9,28] has established that IRF5-IRF4 regulatory axis directs microglial activation towards either pro- or anti-inflammatory polarization. Just like other transcription factors, IRF5 and IRF4 in the cytoplasm may also need to be phosphorylated before they translocate into the nucleus to regulate inflammatory gene expression. Our in vitro assays (Figure 3) quantitatively and morphologically demonstrated the nuclear translocation of both p-IRFs in microglia under ischemic conditions. LPS and IL-4 are established stimulators in vitro for phagocytic pro- and anti-inflammatory response respectively, and widely used in vitro to mimic inflammatory insults in cerebral ischemia in vivo [52,53,54,55]; this was reflected by our IRF translocation data (Figure 3) showing that OGD + LPS induced higher p-IRF5 ratio in the nucleus over cytoplasm than controls, whereas OGD + IL-4 induces higher p-IRF4 ratio. Correspondingly, OGD + LPS and OGD + IL-4 led to higher levels of pro- and anti-inflammatory cytokines respectively (Figure 4) compared to controls. Intriguingly, ND2158 treatment decreased pro-inflammatory cytokine levels to the baseline regardless of the cell stimulation regimen, but lost the effect in the anti-inflammatory cytokine IL-10 when the cells were treated with OGD + IL-4, suggesting a more robust effect of the IRAK4 on microglial pro-inflammatory activation.

IRF5 was reported to be phosphorylated by IRAK4 in macrophages in systemic inflammatory diseases [56]. IRAK4 is a critical component of the TLR signaling pathway that plays a major role in innate immune responses [56,57,58]. It has been found that in monocytes, upon recognition of damage associated molecular patterns (DAMP) by TLR, IRAK4 binds to the death domain region of the adaptor proptein MyD88 and other IRAKs [59,60] to form a complex named Myddosome in the cytoplasm [14,61]. This Myddosome phosphorylates IRF5 to activate the transcription factor, and subsequently facilitates the translocation of p-IRF5 to the nucleus [30,31,62,63,64,65]. As a key component of the Myddosome, IRAK4 itself needs to be activated via phosphorylation of Thr-345, Ser-346, and Thr-342 within its activation loop, with Thr-345 representing the prototypical residue required for the activation [14]. By using ND2158, a specific IRAK4 phosphorylation inhibitor, we found p-IRAK4′s activity not only affects the phosphorylation of IRF5, but also impacts on IRF4, an anti-inflammatory transcription factor that has a counter-effect against IRF5. IRF4 has the biological capacity to compete with IRF5 for binding to the adaptor MyD88 that transmits TLR outside-in signaling for transcription of pro-inflammatory cytokines [16,66,67]. However, it has not been reported before that IRF4 can also be phosphorylated by the IRAK4 comprised Myddosome. Our finding reveals that the competitive binding to MyD88 not only renders IRF4 an antagonistic effect on IRF5 phosphorylation, but also activates IRF4 through the “captured” phosphorylative mechanism.

IRF4 is the predominant transcription factor for microglial anti-inflammatory cytokine production [9]. We noticed that the inhibition of IRF4 phosphorylation by ND2158 had less effect on the IL-10 level when microglia were stimulated by OGD + IL-4 (Figure 4C, lane #4). This phenomenon may suggest that there could be other pathways to phosphorylate IRF4 in addition to the Myddosome. Interleukin-4 (IL-4) signaling regulates IRF4 expression by enhancing the expression and activity of Jumanji domain-containing protein 3 (JMJD3) demethylase, an epigenetic modification procedure mediated by the signal transducer and activator of transcription 6 (STAT6) [10,68]. It has been reported that in T cells, IRF4 was phosphorylated by the serine-threonine kinase Rho-associated, coiled-coil-containing protein kinase 2 (ROCK2), to produce IL-17 and IL-21 [69]. It is likely the same phosphorylation mechanism for IRF4 also exists in microglia. The IRF5-IRF4 regulatory axis established in our previous study is likely responsive to multiple phosphorylation pathways, with IRAK4 signaling predominantly effective on IRF5.

Since ND2158 has a more robust effect on microglial pro-inflammatory cytokine production, it is not surprising that the IRAK4 inhibition confers neuroprotection by increasing MAP2 expression and neuronal viability after OGD + LPS or +IL-4 treatment (Figure 5). Two hour OGD alone did not change the neuronal viability compared with the normoxia control (Figure 5C); however, the OGD exposure did impact on neuronal MAP2 expression and neurite formation (Figure 5D–F), putting neurons in jeopardy. We chose two-hour OGD stimulation for neurons because longer time exposure (e.g., 4 h) has caused extensive neuronal death that has rendered the following experiments unfeasible. However, microglia were subjected to a longer time OGD exposure (4 h) as they are resistant to ischemia. Although the OGD exposure time for microglia and neurons was different in the project due to these reasons, the focus of the study was to mechanistically evaluate the effect of microglial inflammatory responses on neuronal survival in an in vitro world, and our results have shed lights on IRAK4′s role in microglial activation and in neuronal survival after ischemia.

The present study was performed exclusively by in vitro assays to study the phosphorylation and nuclear translocation mechanisms underlying IRF5/IRF4 signaling; therefore, limitations exist that should be kept in mind when interpreting the data. To in vivo manipulate the IRAK4 expression in microglia with transgenic animal models is warranted to elucidate its phosphorylative effects on these IRFs in a “real world”. The Myddosome includes other IRAKs, e.g., IRAK-1 and -2 [14,70]; the contribution of these IRAKs to the phosphorylation of IRFs remains unknown and also warrants further investigation. For most of the experiments, we used SIM-A9 microglial cell line for culture because the commercial cell line yields plentiful cells for various assays. But for the experiments of microglial interaction with neurons (Figure 5) we used primary microglia culture as the neuronal culture was also primary so that both cell types are homologous. Of note, although two regimens of microglia cultures were used, they yielded consistent experimental results, i.e., the IRAK4 inhibition suppressed IRFs’ phosphorylation (Figure 2; SIM-A9 microglia), and impacted on microglial cytokine production to confer neuroprotection (Figure 4 and Figure 5; primary microglia). The phosphorylation and translocation of IRF5/IRF4 were illustrated in the mechanistic diagram (Figure 6).

In summary, the present study explored several key molecular mechanisms for IRF5/IRF4 to regulate microglial response to ischemia. IRAK4 serves as a critical component of the Myddosome including MyD88 that phosphorylates IRF5/IRF4, a procedure that depends on the phosphorylation status of IRAK4 itself. P-IRF5/IRF4 translocate from the cytoplasm to the nucleus after the activation, and regulate the gene expression of inflammatory mediators. Inhibition of IRAK4 activity impacts on microglial pro-inflammatory response predominantly and confers neuroprotection. Since microglial activation is ubiquitous in various neuroinflammatory diseases including stroke, targeting IRAK4 phosphorylation has potential translational value.

## Figures and Tables

**Figure 1 cells-10-00276-f001:**
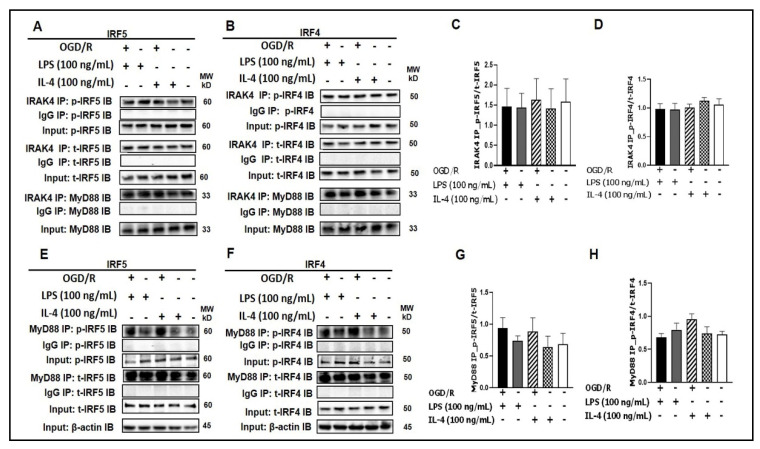
IRF5 and IRF4 bind to MyD88 or IRAK4. (**A**, **B**) SIM-A9 cell homogenates were subjected to Co-IP with anti-IRAK4 antibody followed by immunoblotting for p-IRF5/p-IRF4, t-IRF5/t-IRF4, and MyD88. (**C**, **D**) WB optical density quantification of the ratio of p-IRF5 over t-IRF5 (**C**) and p-IRF4 over t-IRF4 (**D**). (**E**, **F**) SIM-A9 cell homogenates were subjected to Co-IP with anti-MyD88 antibody to detect p-IRF5/p-IRF4 and t-IRF5/t-IRF4. (**G**, **H**) WB optical density quantification of the ratio of p-IRF5 over t-IRF5 (**G**) and p-IRF4 over t-IRF4 (**H**). IgG controls were from the same homogenates of each treatment. *n* = 3 independent experiments/per condition. IB, immunoblot; p, phosphorylated; t, total; IgG, immunoglobulin negative control. One-way ANOVA.

**Figure 2 cells-10-00276-f002:**
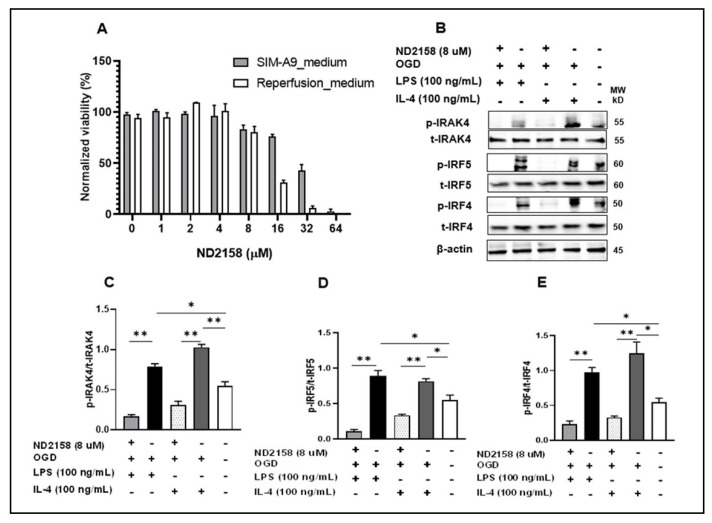
IRAK4 phosphorylates IRF5/IRF4. (**A**) SIM-A9 cells were treated with 8 μM ND2158, chosen by MTS gradient viability assay. (**B**) After treatments, cell homogenates were immunoblotted. ImageJ quantified ratios of p-IRAK4/t-IRAK4 (**C**), p-IRF5/t-IRF5 (**D**), and p-IRF4/t-IRF4 (**E**) in (**B**). Data were obtained from 3 independent experiments. p, phosphorylated; t, total. ** *p* < 0.0001, * *p* < 0.05; One-way ANOVA.

**Figure 3 cells-10-00276-f003:**
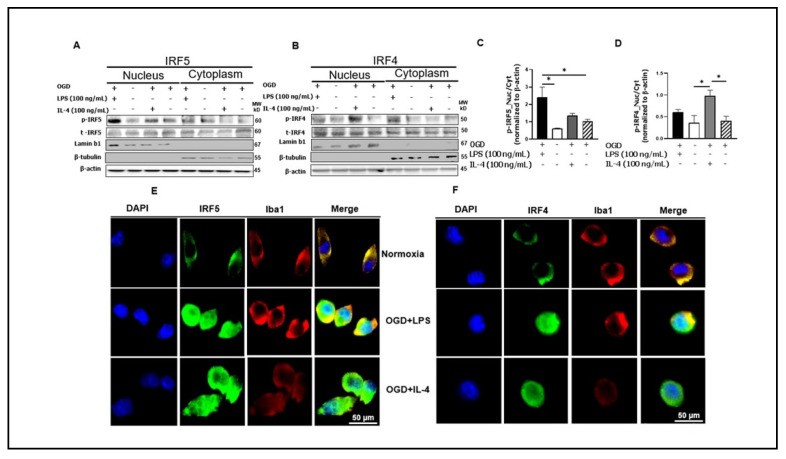
Microglial IRF5 and IRF4 translocate from the cytoplasm to the nucleus. (**A, B**) Cell homogenates were fractionated to nuclear and cytoplasmic portions and Western blot performed in each fraction for p-IRF5/4 and t-IRF5/4. The OD of each band was normalized to β-actin first, and then the quantitative ratio of nuclear/cytoplasmic p-IRF5 (**C**) and nuclear/cytoplasmic p-IRF4 (**D**) were presented. Immunocytochemistry for IRF5 (**E**) and IRF4 (**F**) in microglia showing cytosolic and nuclear expression. Western blot data were from 3 independent experiments. * *p* < 0.05; One-way ANOVA. Lamin b1 and β-tubulin are for the purity validation of nuclear and cytoplasmic fraction respectively. *p*, phosphorylated; *t*, total.

**Figure 4 cells-10-00276-f004:**
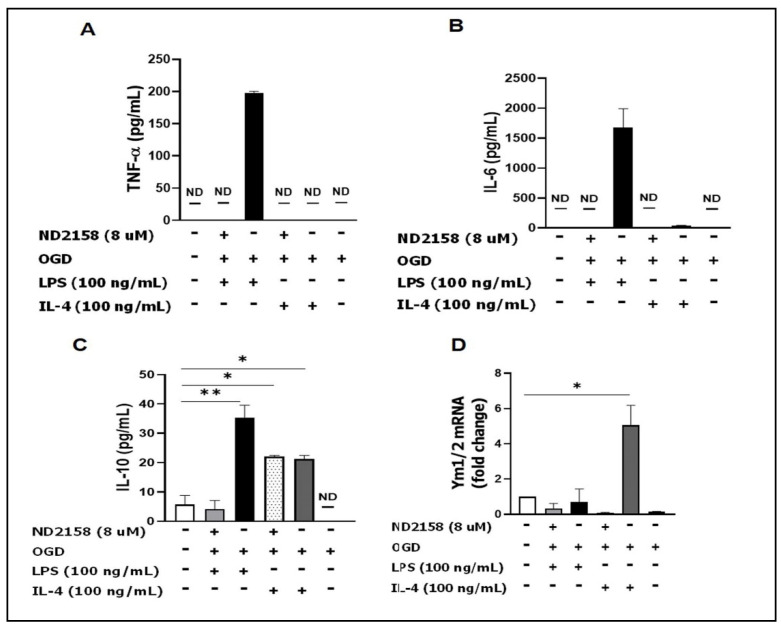
ND2158 treatment decreases microglial cytokine secretion. Primary microglia were cultured with 8 uM ND2158 for 24 h and stimulated with OGD for 4 h and with LPS or IL-4 for 24 h. Cytokines secreted into the culture media were measured by ELISA: (**A**) TNF-α; (**B**) IL-6; (**C**) IL-10. (**D**) Ym1/2 mRNA in cultured cell homogenates. Data were averaged from three independent experiments; ** *p* < 0.0001, * *p* < 0.05; One-way ANOVA.

**Figure 5 cells-10-00276-f005:**
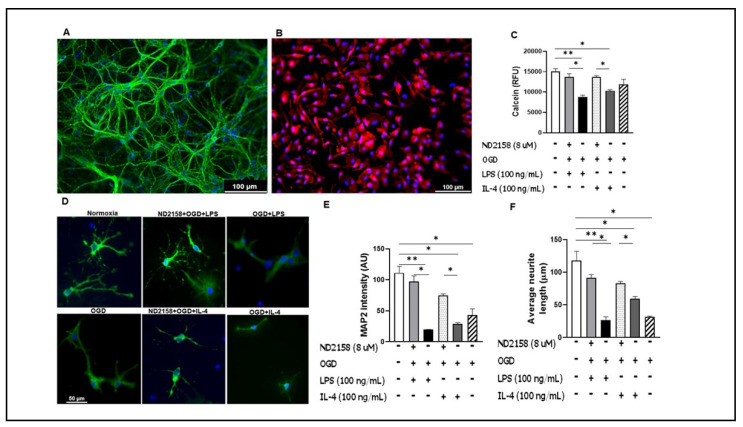
Morphological changes and viability of neurons treated with conditioned medium of microglial culture. Cell culture media from primary microglia treated with ND2158 and stimulated with OGD + LPS or +IL-4, were used to re-perfuse neuronal culture of E16–18 for 24 h. (**A**) Mature neurons stained with DAPI and MAP2 antibody. (**B**) Cultured microglia stained with DAPI and Iba-1. Neuronal morphological changes after exposure of neurons to the conditioned media, and quantification of neurite lengths and MAP2 intensity. (**C**) Neuronal viability quantification with Calcein assay. (**D**) Neuronal morphology after exposure to conditioned microglia culture medium. Quantification of MAP2 fluorescence intensity (**E**) and neurite lengths (**F**) in (**D**). Data were averaged from total 9–12 images in three independent experiments; * *p* < 0.05; One-way ANOVA (**D**); Scale bar = 100 µm (A,B)/50 µm (D).

**Figure 6 cells-10-00276-f006:**
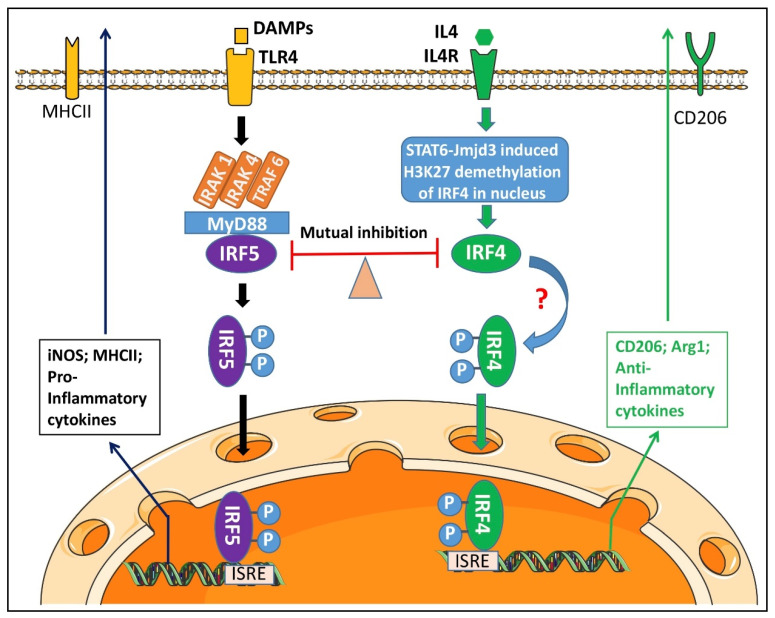
Mechanistic diagram for microglial IRF5/4 activation. Upon DAMPs’ Stimulation, a Myddosome is formed in the cytoplasm to phosphorylate IRF5. P-IRF5 then translocates into the nucleus and bind to ISRE to regulate the expression of pro-inflammatory mediators. IL4 signaling, however, triggers activation of STAT6-Jmjd3-IRF4 pathway. IRF4 competes with IRF5 for MyD88 binding and the IRF5-IRF4 regulatory axis regulates microglial activation in a balanced way.

## Data Availability

The datasets used and/or analyzed in the present study are available from the corresponding author on reasonable request.

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
