# Peer review of "Phosphorylation of Microglial IRF5 and IRF4 by IRAK4 Regulates Inflammatory Responses to Ischemia"

_cells, 2021, doi:10.3390/cells10020276_

Round 1

Reviewer 1 Report

This is a revised manuscript from Ngawa et al that addresses the role of IRAK4 and IRF4/5 in the microglial response to ischemia. It is highly appreciated that the authors addressed many of the reviewers concerns both with new experiments and by editing the manuscript. It is now more convincingly shown that IRAK4 and IRF4/5 play a role in microglial cells under inflammatory conditions.

However, I am afraid, there is still the open question pertaining the role of IRAK4 and IRFs specifically after OGD (ischemia). While the authors have added the OGD alone group to show how microglia react to this treatment when cultured, this does not answer the question what the contribution of OGD is to the IRAK-IRF story. All results presented could be potentially achieved by LPS or IL4 treatment of microglia, without application of OGD. There seems to be some impact, as conditioned media from OGD alone-treated microglia changes neuronal morphology and viability, but this may not be through IRAK4-IRF pathway. To address this, the authors would have to add a ND treatment group with OGD alone to Fig 5 and show that this treatment reverses OGD effects. Otherwise, the title statement “Phosphorylation of Microglial IRF5 and IRF4 by IRAK4 regulates inflammatory responses to ischemia” is not supported.

Author Response

Answer: Thanks for the reviewer's insightful comment. OGD alone has been shown in the revised manuscript that it has minimal effect on p-IRF translocation (Fig. 3) and neuroinflammation (Fig. 4), and it appears to only affect the neuronal morphology but NOT the neuronal viability (Fig. 5C). LPS or IL4 treatment ALONE is not relevant to the topic as this is an ischemic study, and OGD+LPS/IL4 treatments have been widely used in vitro to examine microglial activation after ischemia. Since OGD alone group has minimal effect on neuroinflammation and p-IRF translocation, we think the current data suffice to address the effect of ND2158 even without a third control “ND2158+OGD” in addition to ND2158 and OGD alone controls. We have now clarified more clearly about the OGD alone control in both the result and discussion part.
Happy Holidays!

Reviewer 2 Report

The authors addressed all the comments and revised the manuscript accordingly. 

Author Response

Thanks. 

Reviewer 3 Report

All concerns were clarified by the authors

Author Response

Thanks. 

Round 2

Reviewer 1 Report

As the authors stated in their reply, OGD alone does not have much effect on IRF/IRAK in cultured microglia in vitro. Therefore, it has to be assumed that LPS and IL-4 treatment evoke these responses independently of ischemia. Although LPS and IL-4 alone treatments, as mentioned by the authors, are not directly studied in this work, they do represent necessary controls.

Also, the authors state that OGD+LPS/IL-4 treatments in microglia are widely used as in vitro ischemia models. Please include references for this. LPS is commonly used for pre-conditioning in ischemia but is otherwise as bacterial compound neither produced nor present after ischemia. Therefore, its use in this context appears unusual.

Author Response

Reviewer #1:

As the authors stated in their reply, OGD alone does not have much effect on IRF/IRAK in cultured microglia in vitro. Therefore, it has to be assumed that LPS and IL-4 treatment evoke these responses independently of ischemia. Although LPS and IL-4 alone treatments, as mentioned by the authors, are not directly studied in this work, they do represent necessary controls.
Answer: We apologize for the lack of clarity. In cerebral ischemia, the ischemic insult impacts on neural cells, and resultantly inflammatory mediators (such as CD200, IL-6…etc. playing the similar roles as IL-4 and LPS in vitro) were released from these cells, and in turn act on immune cells (either microglia or peripheral immune cells) to trigger inflammatory responses. The in vivo environment can provide these inflammatory mediators. However, the present work is an in vitro study to mechanistically explore the mechanism underlying microglial activation with microglial culture, which cannot receive signals from those inflammatory mediators. That’s why we use LPS or IL-4 stimulation to mimic the in vivo inflammatory environment. We totally agree with the reviewer that LPS and IL-4 treatment can evoke microglial response independently of ischemia, and that’s exactly what we designed to use LPS and IL-4 to stimulate microglia; otherwise the cultured microglia cannot be activated by the 4-hour OGD. Since we are studying the microglial activation in the context of ischemia, we applied the in vitro ischemia model (i.e. OGD); as shown by our OGD alone control (Fig. 4), the 4-hour OGD alone was not enough to induce significant increase of cytokine levels in microglia. But OGD can prime microglia to make them ready to receive those inflammatory signals as in in vivo ischemia. That being said, LPS or IL-4 alone can also induce microglial activation in vitro without OGD (hundreds of such studies already in
literature [1-3] and the notion has been widely accepted); and a lower does is only
needed if with OGD. Therefore, LPS and IL-4 treatment actually serve as our primary experimental groups, but not controls (it seems unnecessary to set up a “LPS or IL-4” control for themselves, and for a widely-accepted notion). It is absolutely clear that a “LPS or IL-4” control would have the same results as the current study. Again, in the present in vitro study, we used LPS or IL-4 to mimic the in vivo ischemic inflammatory signals, and OGD served as an in vitro ischemia model. LPS and IL-4 have been shown to induce microglial M1 (pro-inflammatory) and M2 (anti-inflammatory) activation respectively [1-3], and in the present study, OGD+LPS or +IL-4 also induced M1/M2 microglial response (Fig.3&4). We have now clarified more clearly about these points in the discussion part.
Also, the authors state that OGD+LPS/IL-4 treatments in microglia are widely used as in vitro ischemia models. Please include references for this. LPS is commonly used for pre-conditioning in ischemia but is otherwise as bacterial
compound neither produced nor present after ischemia. Therefore, its use in this context appears unusual.
Answer: We thank the reviewer for the reminding, and now we have added more
references [4-6] about OGD+LPS and +IL4 treatments for microglial activation (for M1 and M2 activation respectively) in in vitro ischemia in the manuscript. We agree with the reviewer that LPS as a bacterial compound is unlikely to be produced after in vivo ischemia, but other inflammatory mediators are produced in in vivo ischemia that have the similar effects as in vitro LPS. In in vitro studies, LPS has been extensively used to mimic the in vivo inflammatory stimulation for microglia/macrophage studies [2-6], as LPS is a strong inflammatory stimulator that can easily induce the polarization of these phagocytes.